# ON GRADIENT-WEIGHT ALIGNMENT

## ABSTRACT

Evaluating the performance of deep networks against unseen validation data is a crucial step to measure generalization performance. However, ostensibly neither the training nor validation and test data are ever sufficiently extensive to replicate real-world application. This works advocates for a change of perspective for evaluating performance of deep networks. Instead of evaluating against unseen validation data, we propose to rather capture when the model starts to prioritize learning unnecessary or even detrimental specifics of training data instead of general patterns. While this has been challenging to theoretically derive, we propose *gradient-weight alignment* as an empirical metric to determine performance on unseen data from training information alone. Our performance measure is efficient and widely applicable, closely tracking validation accuracy during training. It connects model performance to individual training samples, enabling its use not only for assessing generalization and as an early stopping criterion, but also for offering insights into training dynamics.

## 1 INTRODUCTION

Validation sets are so fundamental and historically tied to machine learning that they tend not to be sufficiently questioned. How well they represent the actual downstream task, especially in critical applications such as medicine, is only seldom a topic in literature. In almost all cases though, validation sets are not sufficiently extensive and there is either a slight distribution shift or even a major change in the downstream application the trained (and validated) model is used for. This begs the question as to what one aims to quantify by using validation sets. On one hand, this is **when** a model does no longer improve its performance on unseen data during training for early stopping and, on the other hand, **how good** the performance on unseen data is estimated to be, *i.e.* the generalization gap. In an ideal scenario, a validation metric should not only measure these two properties but additionally connect model performance to the actual individual data points it was trained on.

Generalization occurs when the representations learned by a model from training data closely align with the true underlying concepts of real-world phenomena. Empirical evidence and theoretical analyses suggest that deep neural networks typically learn broad, simple patterns first, before progressing to more complex, specific details — a process sometimes referred to as a "simplicity bias" (Arpit et al., 2017). Capturing low-complexity patterns first is a training dynamic that is associated with the model improving its ability to generalize effectively to unseen data by capturing the essential features needed for robust predictions. The exact role of learning patterns of increasing complexity or even noise, often associated with memorization/overfitting, is still debated in the context of deep learning (Feldman, 2020). In this work, we empirically show that the *direction of the gradients within the loss landscape spanned by the model weights* allows us to identify whether general patterns are being learned. On a high level, the key intuition underlying our work is as follows: when per-sample gradients seize to be aligned with the model weights during training, this process starts to deteriorate the learned representations in the weights. On the contrary, increasing alignment with the model weights and among gradients indicates improved generalization capabilities allowing for using gradient-weight alignment to predict the performance of the model on unseen data.

**Contributions:** In this work, we propose to leverage the alignment between per-sample gradients and the model weights to efficiently quantify the model performance on unseen data solely based

on information drawn from the training samples and the model while training, which we will from now on refer to as Train-Time Information (TTI). Instead of using possibly non-exhaustive validation datasets with a lack of understanding of how training data affects optimization, our performance measure is intrinsically linked to model convergence itself. Our alignment metric is not only straightforward to compute and applicable even in large-scale settings but also provides insight on a subgroup and even per-sample level. Most importantly, it predicts generalization *without a validation set*, allowing for determining when a model stops learning useful information and for comparing the performance of different models in terms of leveraging available training data information. Our results can be summarized as follows:

- We introduce Gradient-Weight Alignment (GWA) and show how the alignment between per-sample gradients and the model weights corresponds to generalization behavior during different training phases of deep neural networks.
- We propose to use the moments of the alignment scores' distributions to identify these different training phases and the individual samples contributing to the optimization.
- We perform an extensive empirical study to evaluate the predictive capabilities of GWA and show that it does not only measure generalization capability but can also be used as a robust early stopping criterion.

## 2  RELATED WORK

Our work relates to two distinct lines of research in deep neural network optimization focusing either on classifying training or generalization behavior.

To understand *when* a model starts overfitting, our work aims at offering a new perspective on training dynamics, *i.e.* Stochastic Gradient Descent (SGD)'s intrinsic bias to prioritize learning simple, generalizable patterns before shifting towards more complex functions. The idea was first popularized for deep networks by Arpit et al. (2017) and associated with memorization at the end of training. Rahaman et al. (2019) and Kalimeris et al. (2019) showed that this behavior corresponds to an increasing complexity of the model's learned function. SGD initially learns functions characterized by close-to-linear decision boundaries and low frequencies in the Fourier domain. Only later during optimization does SGD lead to non-linear functions of increasing complexity which are less robust to pertubations. This is also reflected in the work by Mangalam & Prabhu (2019), showing that the samples learned early in training can also be correctly predicted by shallow SVMs and Random Forests. While these works focus on the model itself, Refinetti et al. (2023) and Belrose et al. (2024) provide empirical evidence that the simplicity bias of the network function is mirrored by a bias to exploit lower-order input statistics first during training. We adopt the hypothesis of deep networks learning patterns of increasing complexity but shift the focus away from analyzing *when* a model learns *which* features and instead quantify when learning *any* additional feature becomes superfluous for generalization.

Quantifying the generalization gap from TTI, *i.e.* determining *how well* a model performs on unseen data without a validation set, has witnessed much research in recent years with mixed practical value (Jiang et al., 2019). Arguably the most widely used approaches are based on quantifying the curvature of the loss function either by directly computing its Hessian, which tends to be computationally prohibitive, or by approximating it (Hochreiter & Schmidhuber, 1994; Martens & Grosse, 2015; Keskar et al., 2017; Pruthi et al., 2020). However, the curvature of the loss function is most meaningful in the context of stationary points and tends to vary substantially during training while also being sensitive to different choices of hyperparameters (Jastrzebski et al., 2020; Cohen et al., 2021; Gilmer et al., 2022). A key benefit of computing the curvature of the loss function to understand generalization is the ability to do so on a per-sample level, allowing for measuring the influence of individual samples on the optimization process. This connection was first leveraged through the use of *influence functions* for deep neural networks by Koh & Liang (2017), *i.e.* by employing a counterfactual strategy which assesses the impact on a model's output when (hypothetically) omitting a single training sample. An alternative for sample influence that avoids second-order derivatives is *sub-sampling based influence estimation* (Feldman, 2020; Feldman & Zhang, 2020). While both works propose estimators which are computationally demanding and typically computed at the end of training, they not only quantified per-sample influence but also highlighted memorization as a key –yet underexplored– component of generalization.

We contend that, to comprehensively evaluate and characterize model training, we not only need to measure the final generalization gap but should also quantify the influence of individual samples on the optimization *throughout training*. In the following, we explore the interaction between gradients and weights during training and propose a novel approach that allows to connect sample-level information to training behavior and generalization.

## 3 GRADIENT-WEIGHT ALIGNMENT

### 3.1 BACKGROUND

We will motivate GWA by looking into two areas of optimization research: First, we will look into research on *per-sample gradients* and the role of their directions for generalization. And second, we explore how to improve these approaches by evaluating recent theoretical findings on the alignment *between gradients and weights*. Fig. 1 provides a graphical overview of these phenomena.

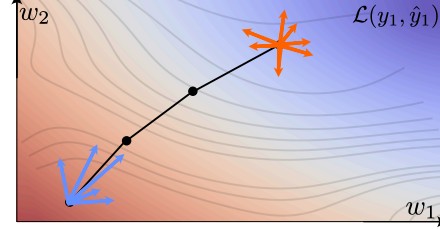

Figure 1: Illustration of varying alignment between per-sample gradients and the model weights during training and corresponding changes in directional dispersion.

The analysis of *per-sample gradient directions* is an area that has only received little interest, while having shown promising initial results and being inherently connected to training behavior of deep neural networks. Intuitively, if there is high alignment between per-sample gradients, the model learns general features prevalent across the dataset. Liu et al. (2020) proposed the *gradient signal-to-noise ratio* across the dataset for each weight to measure the generalization gap. Fort et al. (2020) introduced *stiffness* and experimentally showed that the pairwise per-sample gradients can be used to characterize generalization and class membership. A similar statistic was proposed by Sankararaman et al. (2020) to quantify the convergence rate, with higher alignment between gradients leading to faster convergence. Chatterjee (2020) extends this to the *coherent gradient hypothesis* based on algorithmic stability. All of these approaches are based on the benefit of gradients "pointing the same direction", *i.e. directional alignment* between gradients. However, the aforementioned approaches require the gradients of all samples in the dataset to be stored in memory, rendering their use practically impossible. Moreover, they all aggregate the scores over the dataset, thus not allowing one to draw conclusions on a per-sample basis.

Compared to per-sample gradient alignment, there has been limited empirical research on the alignment *between gradients and weights* for deep networks in practice. Theoretical works have focused on deep homogeneous networks, a wide-ranging class of neural networks allowing for rigorous theoretical study, with their properties having been shown to often extend to other network architectures. If the data can be perfectly classified, the optimization trajectory starts with weights of small magnitude (Glorot & Bengio, 2010; He et al., 2015) which then grows during training, moving away from the origin of initialization and theoretically diverging in norm to infinity (Lyu & Li, 2020). Ji & Telgarsky (2020) show that this behavior is stable, with the weights converging in direction, *i.e.* keeping their orientation constant, and the corresponding gradients aligning with the weights' direction. Recent empirical research has affirmed this stable behavior by analyzing the direction of stochastic batch gradients with respect to a set of optimal weights (Guille-Escuret et al., 2024).

Our work postulates a connecting hypothesis between per-sample gradient directions and the alignment between gradients and weight: we hypothesize that the pairwise alignment between per-sample gradients is reflected in the directional alignment between per-sample gradients and the model weights. By investigating GWA, we cannot just measure the effect of gradient similarity on the optimization process, but also the changing impact of individual samples over time on the direction of the optimization trajectory. We demonstrate that this allows for measuring the two key goals of any validation metric: First, the expected alignment across the dataset allows for estimating the *generalization gap* of the network similar to pairwise per-sample gradient alignment while keeping the ability to trace per-sample contributions. And second, the dispersion of the alignment scores allows for determining *training dynamics* that reflect how well the model is able to capture variance in the dataset and also to detect overfitting.

## 3.2 METHOD

Our method relies on measuring the alignment between per-sample gradients and the model weights. Let $\mathbf{g}_t(x_i) = -\nabla_\mathbf{w}\mathcal{L}(\mathbf{w}_t, x_i)$ denote the negative gradient of the loss function for a single sample $x_i$ with respect to the model weights $\mathbf{w}$ at a time step $t$. We define GWA as the set of alignment scores $\mathcal{A}_t = \{\gamma_t(x_0), \ldots, \gamma_t(x_i)\}$ at a time step $t$, with the per-sample alignment scores being defined as:

$$\gamma_t(x_i) = \cos(\mathbf{g}_t(x_i), \mathbf{w}_t) = \frac{\mathbf{g}_t(x_i) \cdot \mathbf{w}_t}{\|\mathbf{g}_t(x_i)\|\|\mathbf{w}_t\|} . \tag{1}$$

Note that, in theory, if $x_i$ can be perfectly classified, the corresponding alignment score $\gamma_t(x_i) \to 1$ as $t \to \infty$, as shown by Ji & Telgarsky (2020). Although this may never occur in practice, this serves as useful intuition. To understand the model's training dynamics, we are not only interested in individual samples but the behavior of the full dataset as well as quantifying differences within the dataset. Intuitively, the set $\mathcal{A}_t$ can be seen as the directions required to optimally learn each sample in the dataset at a given time step $t$. This set of GWA scores has a bounded probability distribution of values over $[-1, 1]$ with randomness induced by the training data distribution and the stochasticity in model training. [1] Thus, seeing as GWA is a fundamentally distributional quantity, we opt to *leverage the moments of the GWA distribution* to characterize the complex dynamics of the alignment scores during training. We expand on this point next.

**Moments of $\mathcal{A}$**   When looking at the alignment within the dataset, we are on one hand interested in how well the samples are aligned with the weight direction *on average*, as well as the the agreement among the samples, reflected in the *tailedness* of the distribution. Thus, the first and fourth moments, *i.e.* the expectation and the kurtosis, are of particular interest. The expected GWA, $\mathbb{E}[\mathcal{A}_t]$, measures the *directional alignment* of all samples with the model's weights and is representative of the overall (average) direction of the optimization. A large directional alignment indicates a consistent learning direction, which is expected when learning general features during early training phases. Zero or negative directional alignment indicates signal orthogonal or opposing the previous direction of the optimization. To measure the agreement between samples we measure the lack of directional coherence, *i.e.* the *directional dispersion* by computing the excess kurtosis of $\mathcal{A}$, $\mathbb{K}[\mathcal{A}_t]$. The kurtosis reflects how heavy the tail of the alignment score distribution is, with low values being associated with thin tails of the distribution and concentrated per-sample alignment scores. Heavy tails of the GWA distribution indicate high directional dispersion. Note that we use the kurtosis instead of the variance to indicate dispersion, since heavy-tailed distributions frequently occur in practice, and such distributions often do not have a well-defined notion of variance (*i.e.* a second moment). In summary, we will used the term *directional alignment* to denote the expected GWA, and the term *directional dispersion* to denote the excess kurtosis of the GWA distribution.

**Lightweight Estimator**   Computing all alignment scores at each time step $t$ would provide a detailed view of how all individual data points collectively influence the model's training dynamics. However, computing $\gamma_i$ for all samples at every iteration within an epoch is computationally expensive. Instead, we use a mini-batch estimator of $\mathcal{A}_t$ (*i.e.* an empirical estimate of the dataset statistics), and stabilize this estimator by employing an exponential moving average to keep track of the distribution's moments $M_j, j \in \{1, 4\}$ as follows:

$$M[\mathcal{A}_t] = (1 - \alpha) \cdot M[\mathcal{A}_{t-1}] + \alpha M[\mathcal{A}_\mathcal{B}] , \tag{2}$$

with $\mathcal{A}_\mathcal{B}$ being the alignment scores of the current batch and $\alpha$ a small discount factor such as the relative size of the update. The practical computation of gradient-weight alignment is summarized in Algorithm 1, which is lightweight enough to be usable and to be run online even for large-scale models such as Vision Transformers, as seen below. For efficient computation, steps 4-7 can be vectorized, resulting solely in additional memory requirements for the per-sample gradients in a single batch. We empirically found that the aforementioned lightweight estimator correlates excellently with a fully offline computation (using fixed model weights for all samples and computing the score for every gradient individually). Moreover, we experimentally determined that choosing $\alpha = 1/T$ in equation 2, with $T$ denoting the total number of training updates per epoch yields optimal results, and thus use this parameter throughout. In summary, although a full offline computation of every

---

[1]Note that, in the following, we will sometimes abuse notation by re-using the symbol $\mathcal{A}$ to denote both the set and the distribution over the set.

$\gamma_i$ at each $t$ is the optimal (minimum variance unbiased) estimator of the empirical distribution over $\mathcal{A}_t$, the lightweight estimator using mini-batching and stabilized by EMA yields excellent practical results with performance high enough to be run online even for large models. We thus regard the lightweight estimator to be one of the core contributions of our work.

---

**Algorithm 1** Gradient-Weight Alignment (GWA) in SGD

---

**Require:** Number of iterations $T$, batch size $b$, learning rate $\eta_t$, model weights $\mathbf{w}$

1: Initialize $\mathbf{w}_0$
2: **for** each iteration $t = 1, \ldots, T$ **do**
3:      Sample a mini-batch $\mathcal{B}$ of size $b$ from the dataset
4:      **for** each sample $x_i$ in $\mathcal{B}$ **do**
5:          Compute gradient $\mathbf{g}_t(x_i) = \nabla \mathcal{L}(\mathbf{w}_t, x_i)$
6:          Compute per-sample alignment: $\gamma_i = \frac{\mathbf{g}_t(x_i) \cdot \mathbf{w}_t}{\|\mathbf{g}_t(x_i)\| \cdot \|\mathbf{w}_t\|}$
7:      **end for**
8:      Compute directional alignment: $\mathbb{E}[\mathcal{A}_t] = (1 - \sfrac{1}{T}) \cdot \mathbb{E}[\mathcal{A}_{t-1}] + \frac{1}{T \cdot b} \sum_i^b \gamma_i$
9:      Compute directional dispersion: $\mathbb{K}[\mathcal{A}_t] = (1 - \sfrac{1}{T}) \cdot \mathbb{K}[\mathcal{A}_{t-1}] + \frac{1}{T \cdot b} \sum_{i=1}^b \left( \frac{\gamma_i - \bar{\gamma}}{\sigma} \right)^4$
10: **end for**

---

## 4 RESULTS

Recall that the goal of GWA is to provide a metric which quantifies a model's generalization capability, predicts when the model stops learning useful information for generalization, and connects these insights to the training data samples. In the following, we will demonstrate how GWA and the underlying per-sample distribution of alignment scores $\mathcal{A}_t$ provides a solid basis to deriving these insights. We will first show how the distribution and its first and fourth moments, *i.e. directional alignment* and *directional dispersion* evolve during training and connect them to generalization. Finally, we will validate the effectiveness of our method by showing their potential to replace previously established methods and to serve as criteria for gauging generalization capability and for determining early stopping.

### 4.1 DISTRIBUTION OF ALIGNMENT SCORES

To interpret the results below, we note the following. Empirically, generalization is associated with two processes: (1) an increase in di-

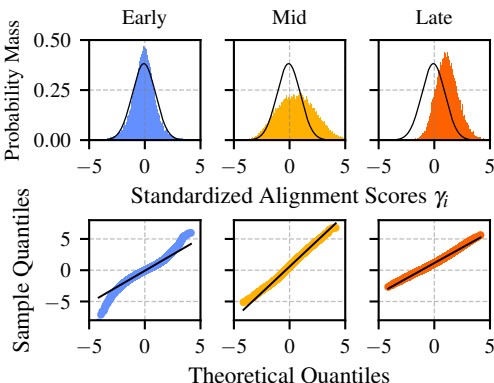

Figure 2: (*Top*) Distribution of alignment scores $\gamma_i$ at different stages during training on CIFAR-100. The distribution is shifted by its mean (Mid) followed by a concentration of the alignment scores (Late) later during training. The corresponding quantile-quantile plots reflect the decreasing kurtosis in the three phases (*bottom*). Gaussian distribution for reference in black.

rectional alignment and (2) a concentration of alignment scores, *i.e.* a decrease in directional dispersion. These two processes can either happen in parallel, which is common on easier tasks and smaller models, or sequentially, with the directional dispersion decreasing *after* the directional alignment has peaked. We hypothesize the increase in directional alignment at the beginning of training to be due to the learning of simple patterns, *i.e.* due to the simplicity bias of gradient descent. In other words, the predominant "signal" during early training is from the general patterns present in the majority of data samples. In later stages, the gradient direction is increasingly influenced by more specific patterns. The samples with such specific patterns represent the tails of the alignment distributions. The heavier these tails are, the stronger the influence of such samples is. Note that this corresponds exactly to the notion of a heavy-tailed distribution (which kurtosis measures): it is not necessarily a large *number* of samples which influence the alignment, but *a small number of very*

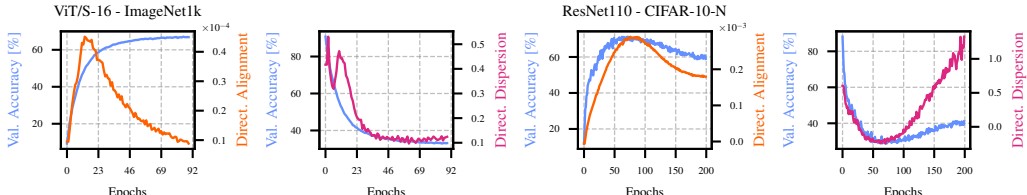

Figure 3: (*Left*) Validation accuracy and corresponding directional alignment and directional dispersion on ImageNet1k trained from scratch following (Steiner et al., 2021). (*Right*) ResNet110 trained on the CIFAR-10-N with human-annotated label noise to provoke overfitting.

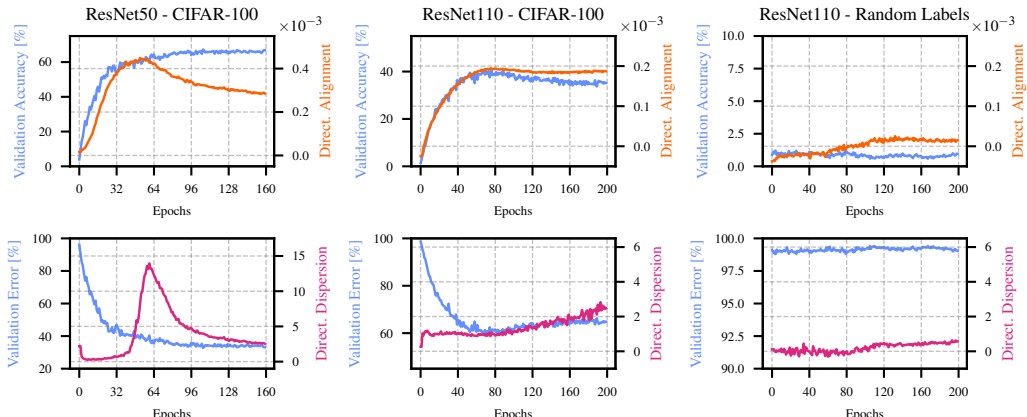

Figure 4: Directional alignment $\mathbb{E}[\mathcal{A}_t]$ and dispersion $\mathbb{K}[\mathcal{A}_t]$ for CIFAR-100 on two models of different size and random labels for comparison. The directional alignment alone is not sufficient to determine generalization behavior but only allows for deriving training behavior in tandem with the kurtosis of $\mathcal{A}_t$. Training a model on random labels equivalent to (Zhang et al., 2017) leads to no substantial change in neither directional alignment nor directional dispersion, staying around 0.

*influential samples*. An increasing directional dispersion in mid training thus reflects variation in the dataset and the model's capability to capture it. In other words, if a model has sufficient learning capacity, the directional dispersion will tend to decrease towards the end of training. In contrast, a poorly generalizing model (*i.e.* an overfit model) will have worse directional alignment during the initial generalization phase and increasing directional dispersion in later stages.

Fig. 3 shows this behavior exemplarily on two datasets: In the left subplots, we trained a ViT/S-16 on ImageNet and observe an increase in directional alignment in early training together with an increase in directional dispersion followed by a decrease in late training. This pattern is a clear sign of generalization. In contrast, the right subplots show a ResNet110 trained on CIFAR-10 with human-annotated label noise to provoke overfitting. While the model generalizes well initially, the label noise leads to overfitting in later stages indicated by a decreasing directional alignment and more importantly an increase in directional dispersion. Similar behavior can be seen for models of different sizes trained on CIFAR-100 in Fig. 4. Note that ResNet110 has *fewer* parameters than ResNet50, and thus lower learning capacity. Therefore, after ca. epoch 100 the directional dispersion begins to increase, corresponding to a decrease in test accuracy. Finally, the most drastic example is observed in the right subplots, where the model is trained on randomized labels. Here, the directional alignment remains negative throughout most of the training and stabilizes around 0. We thus conclude that this model is unable to learn generalizable patterns, which is mirrored in a test error of 99% throughout training.

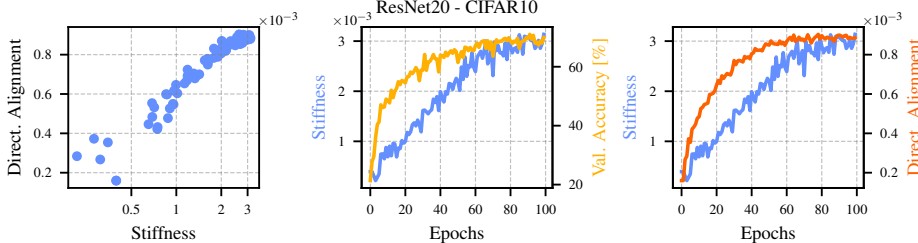

Figure 5: (*Left*) Near-perfect log-correlation of pairwise per-sample gradient alignment (*stiffness* Fort et al. (2020)) and our directional alignment $\mathbb{E}[\mathcal{A}_t]$. Development of stiffness during training versus validation accuracy (*center*) and directional alignment (*right*). Note that directional alignment tracks the validation accuracy closely (yellow and red curves).

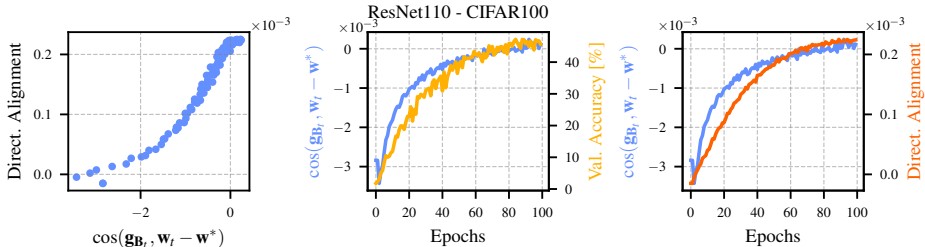

Figure 6: (*Left*) Correlation between alignment of batch gradient $\mathbf{g}_t \mathcal{B}$ and optimal set of weights $\mathbf{w}^*$ as proposed by Guille-Escuret et al. (2024) and our directional alignment $\mathbb{E}[\mathcal{A}_t]$. Development of the alignment with regards to $\mathbf{w}^*$ during training versus validation accuracy (*center*) and directional alignment (*right*). Note that directional alignment tracks the validation accuracy near-perfectly (yellow and red curves).

## 4.2 GWA vs. Methods from Prior Work

There are two related works particularly relevant to our approach. First, the research on gradient alignment introduced in 3.1 provides a direct connection to the convergence rate of optimization and model generalization. We compare against the expected pairwise gradient alignment as introduced in Fort et al. (2020); Sankararaman et al. (2020), in particular the definition of *stiffness* as $\mathbb{E}_{i \neq j}[\cos(\mathbf{g}_t(x_i), \mathbf{g}_t(x_j))]$. Intuitively, our proposed directional alignment can be seen as the alignment between gradients with a reference vector, specifically the weights $\mathbf{w}_t$, rather than the pairwise alignment between gradients. This renders our proposed quantity much more memory-efficient, as it does not require holding all gradients in memory to compute the pairwise scores, while allowing for directly tracing the per-sample contributions to the expectation of the alignment distribution $\mathcal{A}_t$. Fig. 5 shows that directional alignment not only measures the same training dynamics as stiffness, but actually tracks the corresponding validation accuracy more closely throughout training.

While GWA functions similarly to Ji & Telgarsky (2020) by considering the current set of weights $\mathbf{w}_t$ to compute the alignment scores $\gamma_i$, we are also interested in how well the gradients point towards the *optimal* set of weights $\mathbf{w}^*$. Guille-Escuret et al. (2024) introduced the alignment between the stochastic batch gradient $\mathbf{g}_t(\mathcal{B})$ and the vector pointing towards the optimum $\mathbf{w}^*$ from the current set of weights $\mathbf{w}_t$ as a ratio between loss curvature and the error bound during optimization. However this is prohibitive to compute in practice as the true optimum is unknown and approximating it requires to re-run the optimization at least once to find an optimal set of weights $\mathbf{w}^*$ for a given run. Even then, this optimum is not guaranteed to be global, and thus, multiple repetitions would theoretically be required. Fig. 6 shows that directional alignment, while not measuring the same quantity, is predictive of the same relative behavior during optimization indicating a relationship between GWA and the loss curvature relative to the distance to the optimum. Not only is directional alignment independent of $\mathbf{w}^*$, it also traces the change in validation performance more closely.

In summary, directional alignment measures training behavior that has been established by prior works on pairwise per-sample gradient alignment and by analyzing the loss curvature with respect to the optimum while requiring neither the memory nor the compute overhead of the aforementioned approaches. Additionally, GWA inherently allows for tracing how individual samples contribute to the measured score, contrary to the other approaches, which aggregate over the dataset. We leverage this property in the next section.

### 4.3 MEASURING SAMPLE MEMORIZATION

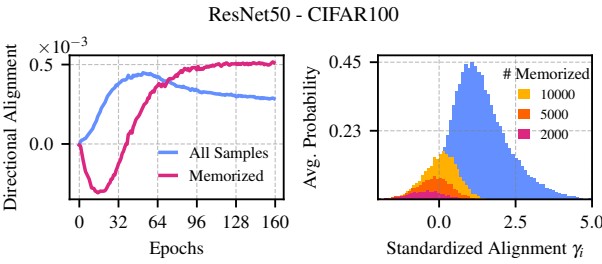

The GWA distribution of alignment scores $\mathcal{A}_t$ is composed of per-sample values. This allows for analyzing contributions of individual samples to directional alignment and dispersion. This poses the question as to *which samples are not aligned* with the expectation of the distribution and whether these samples change their gradient direction throughout training. Despite the inherent complexity of quantifying sample influence on optimization, we focus on two recent areas of research to demonstrate the effectiveness of GWA: First, we evaluate memorization of samples by the model by analyzing the behavior of *highly memorized* samples throughout training. We use the approach and

Figure 7: (*Left*) Memorized samples can be identified by negative alignment during initial training phase, whereas memorization with positive alignment sets in after reaching maximum directional alignment. (*Right*) Average directional alignment until $\max_t \mathbb{E}[\mathcal{A}_t]$ for the different number of most memorized samples compared to overall GWA distribution $\mathcal{A}_t$. Higher prevalence in the negative tail of $\mathcal{A}_t$.

the corresponding memorization scores proposed by Feldman & Zhang (2020) for CIFAR-10 trained on a ResNet50 for this purpose. Here, a sample has high memorization if excluding the sample during training leads to a large change in accuracy for this specific sample. Fig. 7 (*left*) shows that the 2 000 samples with the highest memorization score are opposing the overall directional alignment during in the initial training epochs. This corroborates the simplicity bias, whereas the optimization focuses on simple, highly prevalent patterns first. The distribution of alignment scores $\mathcal{A}_t$ (Fig. 7 *right*) shows a similar picture: 2 000 of the most memorized samples are responsible for the most negative part of the alignment distribution. This precisely aligns with the notion of a heavy-tailed distribution where few samples have an outsize effect. Note moreover that in Fig. 7 (*left*), the highly memorized samples become increasingly aligned only *after the overall alignment has peaked*. As shown by Feldman (2020), further improvements in generalization performance later during training are mostly driven by memorized samples which exhibit the highest directional alignment, indicating the necessity of memorization for improving generalization in learning tasks.

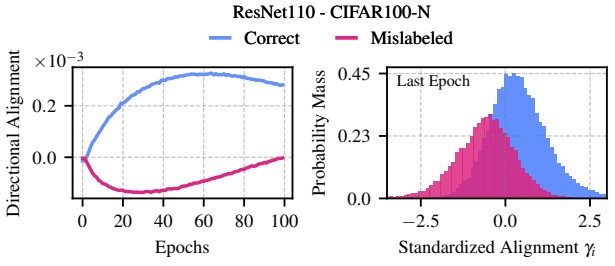

While memorization can be potentially beneficial for generalization, we expect label noise to have a detrimental effect on model performance. In the following, we will repeatedly make use of the CIFAR-N (Wei et al., 2022) dataset which offers versions of CIFAR-10/100 with different levels of human-annotated label noise allowing us to evaluate varying model performance and overfitting. We train the ResNet110 from Fig. 4 on CIFAR-N with $40\%$ label noise, *i.e.* a large part of the samples are mislabeled by human annotators to a class with high similarity. Observing Fig. 8, it can be seen that the net-

Figure 8: (*Left*) The mislabeled samples of CIFAR-100-N are negatively aligned throughout the training, potentially opposing the direction of correctly classified samples. (*Right*) Mean shift between correct and mislabeled samples in $\mathcal{A}_t$ at the last epoch.

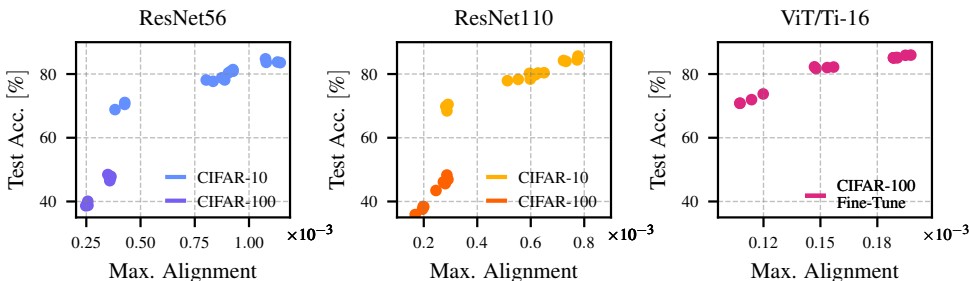

Figure 9: Correlation between test accuracy and the maximum directional alignment $\max_t (\mathbb{E}[\mathcal{A}_t])$ during training on CIFAR-10-100-N with different levels of label noise. The ViT/Ti-16 is fine-tuned on ImageNet21k weights (Dosovitskiy et al., 2021).

work is still able to pick up signal with an on-average positive directional alignment. While there is a substantial overlap between the correct and mislabeled samples in the alignment distribution $\mathcal{A}_t$, the average directional alignment of the mislabeled samples stays below zero, *i.e.* opposing the weight direction, throughout the whole optimization. Notably, this is unlike the highly memorized samples in the previous experiment which were learned by the model at some point during training. To summarize, GWA helps to better understand the complex patterns of sample influence on the model performance. We showed that memorized samples only get learned later during training (as also shown *e.g.* in Stephenson et al. (2021)), while label noise on average tends to oppose the directional alignment of the entire dataset.

### 4.4    Assessing the Generalization Gap

Predicting generalization from TTI alone has been a major challenge in research. Here we show that this is to some extent possible with GWA. Directional alignment and directional dispersion have shown great promise in tracing test accuracy throughout training in our previous experiments. To quantitatively evaluate if it is possible to use the alignment scores to measure generalization performance, we train models of different sizes from scratch and with fine-tuning. We use the previously introduced CIFAR-N dataset, which includes versions of CIFAR-10 and CIFAR-100 with different levels of human-annotated label noise, and evaluate on the duplicate-free ciFAIR (Barz & Denzler, 2019) test set in a large-scale experiment over varying generalization performances with and without overfitting. This prevents the model from achieving artificially high test accuracy due to training data being near-identically duplicated in the test set.

We first use the maximum directional alignment $\max_t (\mathbb{E}[\mathcal{A}_t])$ during training and compare it against the maximum accuracy on ciFAIR. We choose directional alignment without including dispersion as it has shown strong correlation with the expected pairwise gradient alignment in Sec. 4.2, which has been shown in previous works to relate to generalization. Fig. 9 shows a clear correlation within each dataset and model architecture, with lower test accuracy being associated with lower maximum directional alignment. Notably, this correlation between directional alignment and generalization is maintained on the pre-trained ViT, indicating that the alignment scores do not only work when training from scratch on randomly initialized weights, but also for pre-trained models.

The strong correlation with test accuracy poses the question if we can use directional alignment as an early stopping criterion. We take the time step with the lowest directional dispersion $\arg\min_t (\mathbb{K}[\mathcal{A}_t])$ subject to previously having reached the peak directional alignment. We compare our GWA-based early stopping against standard early stopping based on the highest validation accuracy. Table 1 shows that using GWA to determine the optimal epoch returns model performance that comes close to using the validation accuracy for early stopping while using only the available information during training. This is even the case under label noise of up to $40\%$. Additionally, we assess whether the difference between the two early stopping criteria decreases when evaluated on test data subjected to realistic perturbations and corruptions, as encountered in practical applications. For this purpose, we utilize the CIFAR-C and CIFAR-P test sets from Hendrycks & Dietterich (2019), which consist of various versions with differing levels of alterations to the original test sets,

| | Noise | Test Accuracy [%] | | CIFAR-C [%] | | CIFAR-P [%] | |
|---|---|---|---|---|---|---|---|
| | | GWA-Stop | Val.-Stop | GWA | Val. | GWA | Val. |
| CIFAR-10 | – | 79.59±1.04 | 81.81±0.54 | 66.08 | 66.45 | 68.76 | 68.89 |
| | 9% | 77.34±0.79 | 78.73±0.59 | 62.33 | 63.11 | 64.27 | 64.91 |
| | 17% | 75.95±0.37 | 77.03±0.55 | **59.27** | 56.86 | **61.43** | 58.78 |
| | 40% | 68.40±1.37 | 70.01±0.17 | 55.56 | 57.82 | 56.20 | 57.90 |
| CIFAR-100 | – | 41.04±0.30 | 44.14±1.12 | 26.29 | 27.49 | 26.61 | 27.49 |
| | 20% | 40.39±0.60 | 41.13±0.74 | 22.51 | 22.88 | 22.37 | 22.62 |
| | 40% | 36.63±0.43 | 36.75±0.29 | **22.96** | 21.69 | **22.75** | 21.36 |

Table 1: Test Accuracy for early stopping with either GWA or based on validation accuracy for CIFAR-10 and CIFAR-100 on a ResNet56 with different levels of human-annotated label noise from CIFAR-N. Additional average test accuracy across different corrupted (*CIFAR-C*) and perturbed (*CIFAR-P*) test sets from Hendrycks & Dietterich (2019) to evaluate robustness of the model.

designed to evaluate the robustness of models. On these more or less out-of-distribution test sets, GWA sometimes *even outperforms validation accuracy in determining the optimal epoch for test performance*. Thus, early stopping based on GWA not only provides a method that relies solely on TTI but also excels in identifying robust models if required for the downstream tasks. This makes it particularly advantageous in fields with limited data and significant domain shifts in application, such as the medical field.

To summarize, directional alignment not only correlates with test performance when evaluating the performance of a model for a given dataset by using TTI only but can also be used as an early stopping criterion.

## 5 DISCUSSION AND CONCLUSION

The reliance on per-sample gradients is a bottleneck for GWA required if we want to trace information back to individual samples. With the recent progress in improving efficiency in this area due vectorization and compilation, we are confident that computing per-sample gradients will not only get even more feasible but also widely adapted in practice. Even with further improvements in efficiency, validation accuracy itself will not be replaced by GWA due to its simplicity and reliability in most cases. Nontheless, GWA is a expressive metric that will be especially helpful when additional information about the training is required, domain shifts are to expected in application, and/or only little data is available. Similarly, directional alignment and dispersion are just two moments of the GWA distribution. Analyzing time dynamics of the alignment score distribution has the potential to further enhance current results and presents a promising avenue for future research. An interesting aspect of the gradient-weight alignment scores is their small magnitude. Due to the high representation dimensionality of gradients and weight vectors of deep networks, the cosine similarity of the per-sample alignment scores tends to get smaller for larger models, making comparisons across model architectures challenging. We consider solving this an interesting direction for future work.

In summary, we investigated GWA and introduced directional alignment and directional dispersion as two metrics to capture training dynamics during optimization. We demonstrated that both methods not only track validation accuracy throughout training but can also efficiently approximate other metrics, such as the expected pairwise gradient alignment. Their ability to trace per-sample influence through memorization and mislabeling, estimate generalization performance, and serve as a robust early stopping criterion makes GWA a promising validation measure. We hope that this work will inspire renewed interest in unbiased performance metrics and further exploration of the relationship between individual data samples and model training dynamics.

## 6    REPRODUCIBILITY STATEMENT

To ensure the reproducibility of our work, we made several key prior decisions. Our research exclusively utilizes publicly available datasets that are widely recognized in the literature. Additionally, the models trained during our evaluations either adhere to standard training setups (*e.g.* , *ViT/Ti-16 on ImageNet1k or ResNet50 for memorization on CIFAR-100*) or are commonly used models for the task (*e.g.* , *ResNet56 and ResNet110 on CIFAR-10-100-N*). Notably, due to the nature of per-sample gradients we adapted all ResNets to use group normalization instead of the commonly applied batch normalization. For easier reproduction, code is available in `JAX` at: https://anonymous.4open.science/r/iclr-73F2

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
