# OpenReview forum: "On Gradient-Weight Alignment"
_ICLR.cc/2025/Conference — Submitted to ICLR 2025_

### Official Review · Reviewer_383N · 2024-10-30

**Soundness:** 1
**Presentation:** 2
**Contribution:** 1
**Rating:** 1
**Confidence:** 5

**Summary:**

For each sample $x$ and training step $t$, the paper considers the alignment between the negative sample gradient ${\bf g}_t (x)$ of the loss function corresponding to the sample $x$ and the model weight vector ${\bf w}_t$ at the training step $t$ and defines a notion called per-sample alignment score $\gamma_t (x)$ as the correlation between  ${\bf g}_t (x)$ and ${\bf w}_t$. Since $x$ can be regarded as a random variable, so is  the per-sample alignment score $\gamma_t (x)$. The paper then refers to the first moment of the random variable  $\gamma_t (x)$ as the directional alignment, and the kurtosis of the random variable  $\gamma_t (x)$ as the directional dispersion. Both the directional alignment and dispersion are estimated from a mini batch during the course of training via the standard exponential moving average approach. The estimated directional alignment and dispersion are then used to analyze empirically the training dynamics during the course of training. Based on limited experiments, some observations are made in terms of using directional alignment and dispersion to track validation accuracy during the course of training, determine an early stopping time without using any information from the validation set, and analyze performance contributions from individual samples.

**Strengths:**

The only strength I can think of is the introduction of per-sample alignment score $\gamma_t (x)$ defined as the correlation between  ${\bf g}_t (x)$ and ${\bf w}_t$.

**Weaknesses:**

The paper is not mature enough. Its position is not clear, experiments are limited, research statements are inconclusive, and no new insights are really provided.

The authors have to think harder on how to position the paper. As a suggestion, a promising position is to investigate how to use the estimated directional alignment and dispersion to determine an early stopping time without relying on any information from validation sets so that the trained DNN is more robust with respect to different validation sets in the downstream applications. The results presented for this direction so far are limited and not convincing enough. The authors are encouraged to continue along this direction. Other observations and discussions are vague and inconclusive, and will not go anywhere.

**Questions:**

1. The explanations and discussions near the bottom of Page 5 are not new. They are well-known and can be explained from the values of cross entropy loss.

2. Something is wrong in Figure 3. For each model in Figure 3, the blue curve on the right sub-figure seems to be validation error rate, not validation accuracy. In addition, why not put the three curves (the validation accuracy curve, directional alignment curve, and directional dispersion curve) into one sub-figure?

3. The observations from Figures 3 and 4 are not conclusive, and vague.

4. Contradicting to the statement made in Lines -3 and -2 on Page 6 (bottom of Page 6), the rightmost sub-figure in Figure 4 does not show any negative directional alignment.

5. Figures 3 and 4 contradict to Figure 5. Putting three together, one cannot conclude that the directional alignment curve tracks the validation accuracy closely.

---

> ### Author Response · Authors · 2024-11-20
>
> Thank you for your review of our submission and your feedback.
>
> Your criticism is voiced assertively. While you raise some valid points which we are grateful for, we feel that we have to push back on a few aspects.
>
> > “[...] a promising position is to investigate how to use the estimated directional alignment and dispersion to determine an early stopping time without relying on any information from validation sets so that the trained DNN is more robust with respect to different validation sets in the downstream applications. The results presented for this direction so far are limited and not convincing enough. The authors are encouraged to continue along this direction.”
>
> We agree that this proposed direction is of particular interest and a key motivator of our work. However, the purpose of our work is not solely to introduce a novel early stopping criterion but to find a measure that inherently links prediction performance to the training samples used during optimization. We thus see the evaluation of sample memorization and sample mislabeling in Section 4.2 and 4.3 as crucial for the reader to understand what GWA is measuring and how to better understand the metric for use in practice.
>
> > “[...] in Figure 3, the blue curve on the right sub-figure seems to be validation error rate, not validation accuracy. In addition, why not put the three curves (the validation accuracy curve, directional alignment curve, and directional dispersion curve) into one sub-figure?”
>
> Thanks for catching the mistake, we will fix it. Regarding combining different figures, we have considered this but kept them separate for improved clarity.
>
> > “The observations from Figures 3 and 4 are not conclusive, and vague.”
>
> The figures show the behavior described in line 258 and following, which is characterized by an increase in directional alignment when learning general features initially, and either a decrease or increase of directional dispersion after reaching peak alignment (with the later being associated with overfitting). As this is consistent across all experiments we would be happy to better understand which observations you are referring to.
>
> > “Contradicting to the statement made in Lines -3 and -2 on Page 6 (bottom of Page 6), the rightmost sub-figure in Figure 4 does not show any negative directional alignment.“
>
> We agree that this is not worded properly and will update this. The focus should be on both directional alignment and dispersion staying consistently around 0 without showing any sign of learning useful information.
>
> > “Figures 3 and 4 contradict to Figure 5. Putting three together, one cannot conclude that the directional alignment curve tracks the validation accuracy closely.”
>
> The main finding of Figure 5 is not how well directional alignment tracks validation accuracy but how well it is correlated to stiffness and how the two change during training. It thus does not contradict Figure 3 and 4, in fact, Figure 5 shows similar behavior to what we can observe e.g. in Figure 3 right. We will still remove the last phrase in the caption to avoid any confusion here.

---

> ### Comment · Reviewer_383N · 2024-11-28
>
> Thank you for your responses. My concerns expressed in the weakness section remain the same. It is easy for the reader to understand your proposed metrics: the directional alignment and dispersion. The issue, however, lies in how to utilize the proposed metrics to derive something good, which is lacking at this point. I hope my suggestion can help you improve your future position and research along this line.

---

### Official Review · Reviewer_dUVR · 2024-11-03

**Soundness:** 3
**Presentation:** 3
**Contribution:** 2
**Rating:** 5
**Confidence:** 4

**Summary:**

The paper proposes a novel approach to evaluating the performance of deep neural networks without relying on unseen validation data. Recognizing that training, validation, and test datasets are often insufficient to replicate real-world applications, the authors advocate for a shift in perspective. Instead of traditional validation methods, they introduce gradient-weight alignment as an empirical metric to determine a model's generalization performance using only training data.

This metric identifies when a model prioritises learning unnecessary or detrimental specifics of the training data rather than capturing general patterns. The proposed method is efficient and widely applicable, mirroring validation accuracy during training. Connecting model performance to individual training samples not only aids in assessing generalization and serves as an early stopping criterion but also offers valuable insights into the training dynamics of deep networks.

**Strengths:**

1. The method proposed in this paper is interesting and well-motivated, which allows the evaluation of model generalization without using a validation dataset.
2. The method is efficient and allows performance at each time step during training, which allows the model to stop training before overfitting occurs on general unseen data.
3. The author shows extensive empirical results that reveal the correlation between proposed metrics and model performance on different image-classification tasks.

**Weaknesses:**

The paper provides many evaluations on C10 and C100 image classification tasks, which are pretty narrow analyses for evaluation generalization purposes; more experiments on ImageNet-1k would be beneficial.

**Questions:**

1. Based on the abovementioned weakness, will these metrics work on other tasks, such as object detection or segmentation?
2. Many recent works on Zero-cost NAS metrics also mention training dynamics, such as:
[1] Li, G., Yang, Y., Bhardwaj, K., & Marculescu, R. ZiCo: Zero-shot NAS via inverse Coefficient of Variation on Gradients. In The Eleventh International Conference on Learning Representations.
[2] Xiang, L., Hunter, R., Xu, M., Dudziak, Ł., & Wen, H. (2023, December). Exploiting network compressibility and topology in zero-cost NAS. In International Conference on Automated Machine Learning (pp. 18-1). PMLR.

How do the proposed methods perform differently with those works? It would be good to compare the performance prediction ability by showing the correlation on specific NAS benchmarks, like NASbench101, 201 or NASlib, that might be a good approach to show that this metrics is well correlated to the trained validation accuracy.

---

> ### Author Response · Authors · 2024-11-20
>
> Thank you for your reviewing and the feedback for our submission.
>
> > “The paper provides many evaluations on C10 and C100 image classification tasks, which are pretty narrow analyses for evaluation generalization purposes; more experiments on ImageNet-1k would be beneficial.”
>
> We focused on CIFAR-10/100 as we can use versions of the same dataset for showing a variety of behaviors (from compute intensive related work such as stiffness, to using memorization scores from Zhang et al., and labeling errors in CIFAR-N). However, we fully agree that more experiments on ImageNet-1k will strengthen our findings and will extend our analysis in an updated version, especially for Table 1.
>
> > “[...] will these metrics work on other tasks, such as object detection or segmentation.”
>
> Yes, as long as we are able to compute a form of per-sample gradients, the method is applicable. Do you have any specific datasets in mind that you would be interested to see / that you think could be challenging?
>
> > “How do the proposed methods perform differently with those works? It would be good to compare the performance prediction ability by showing the correlation on specific NAS benchmarks [...].”
>
> The main difference is that our metric is not zero-shot and still requires training the network - at least for a few epochs. This does seem like an interesting application though, and we will look into using GWA in this domain.

---

> > ### Comment · Reviewer_dUVR · 2024-11-21
> >
> > Thanks to the author's response, the COCO dataset seems good for object detection tasks. I still think the GWA method is interesting, yet more experimental results may be needed to evaluate its effectiveness fully.

---

### Official Review · Reviewer_Marc · 2024-11-03

**Soundness:** 2
**Presentation:** 3
**Contribution:** 2
**Rating:** 3
**Confidence:** 3

**Summary:**

This paper proposes a novel approach to assessing model generalization by focusing on the alignment between model gradients and weights during training. Specifically, it introduces a metric termed Gradient-Weight Alignment (GWA). Motivated by the limitations of validation sets in representing real-world distributions, the authors seek a method to evaluate model performance using only training data. The proposed algorithm calculates the cosine similarity between per-sample gradients and model weights. This implementation involves monitoring directional alignment and directional dispersion as two key indicators. The authors claim that high alignment and low dispersion correlate with effective generalization. Experimentally, GWA is shown to closely track validation accuracy across various models and datasets, and it serves as a robust early stopping criterion—especially useful in scenarios with label noise or significant domain shifts.

**Strengths:**

1. This paper introduces a lightweight and scalable estimator for alignment scores that is applicable to large models and noisy datasets. The two proposed indicators can be used to predict generalization performance.
2. The proposed metric is easy to implement and expected to be more stable than previous methods.

**Weaknesses:**

1. The main concern is the significance of this work. The idea of measuring gradient similarity is not new. Although this paper considers the alignment between gradients and weights, which is different from existing methods, it’s not clear why the former is superior to the latter in terms of generalization estimation. Regarding memory cost and computational complexity, I think some existing methods, such as Stiffness, can also be extended to a faster stochastic version, similar to the operation in Algorithm 1, line 9. Apart from empirical evaluations, it would be better to see more insightful analyses of GWA’s effectiveness.
2. For the experiment assessing the generalization gap in Section 4.4, only a validation set is used as the baseline. Other popular metrics that do not rely on a validation set should also be evaluated. Additionally, the reported accuracies on both CIFAR-10 and CIFAR-100 are too low. Popular deep models, such as ResNet-18, typically achieve >90% accuracy on CIFAR-10 and >70% on CIFAR-100 with standard training techniques. The current experiments do not sufficiently demonstrate that GWA achieves state-of-the-art performance in predicting the generalization gap.
3. The experiments in Sections 4.1 and 4.2 are mostly qualitative, showing the correlation between directional alignment and validation accuracy. This could be evaluated in a more rigorous way, such as through quantitative analyses comparing model selection using GWA with baseline algorithms.
4. Minor Issues:
   - 1) The y-axis of the second and fourth plots in Figure 3 should be labeled *Val. error* instead of *Val. acc.*
   - 2) The discussion in lines 412-416 seems unclear, as alignment is naturally easier when only a subset is used to train the model (making fitting easier).
   - 3) The second and third paragraphs in Section 4.4 lack clear logic. They initially suggest that directional alignment is a good predictor of generalization but then recommend using directional dispersion rather than alignment to decide when to stop training.

**Questions:**

1. In Figure 7 (right), do you choose a specific time/epoch t or calculate the average over the entire training process?
2. In Table 1, what is the size of the validation set?

---

> ### Author Response · Authors · 2024-11-20
>
> Thank you for your feedback on our submission.
>
> > “The idea of measuring gradient similarity is not new. Although this paper considers the alignment between gradients and weights, which is different from existing methods, it’s not clear why the former is superior to the latter in terms of generalization estimation. Regarding memory cost and computational complexity, I think some existing methods, such as Stiffness, can also be extended to a faster stochastic version, similar to the operation in Algorithm 1, line 9.”
>
> While the idea of gradient similarity measures is not new - in fact it is the base of our work (see line 126) - these measures are so far not widely used outside the respective papers introducing them. This is mainly due to the substantial compute overhead associated with these former approaches. And while we would be happy to see extensions of previous methods with our proposed estimator, computing alignment scores based on pairwise similarity in comparison to a reference vector, such as the weights, will inherently always require more compute and introduce additional stochasticity.
>
> > “Apart from empirical evaluations, it would be better to see more insightful analyses of GWA’s effectiveness.”
>
> We would be grateful if you could elaborate more on what you mean by measuring the effectiveness of GWA.
>
> > “[...] only a validation set is used as the baseline. Other popular metrics that do not rely on a validation set should also be evaluated. Additionally, the reported accuracies on both CIFAR-10 and CIFAR-100 are too low.”
>
> Do you have any particular metrics in mind here? As mentioned in line 97, we consider sharpness as one of the most popular alternatives. However, not only does it have shortcomings discussed in Section 2 but we also do not see any additional benefit to using it in comparison to the more informative validation accuracy. We would be happy if you could elaborate more on the potential benefit here.
>
> Regarding the accuracies on CIFAR-10/100, this is only for the ResNet-based models which need to be adapted to use group normalization in comparison to batch norm to compute per-sample gradients. We will look into changing these with inherently batch-free methods in an updated version.
>
> > “This could be evaluated in a more rigorous way, such as through quantitative analyses comparing model selection using GWA with baseline algorithms.”
>
> Are there any specific experiments you would be interested to see? In particular delimited from the results in Table 1.
>
> > “The discussion in lines 412-416 seems unclear, as alignment is naturally easier when only a subset is used to train the model (making fitting easier).”
>
> Could you please elaborate on this point? We use the whole dataset throughout training, only the model is influenced by different subsets of that dataset throughout training as part of the intrinsic bias introduced by SGD. In addition, we are not sure why alignment should be easier on a subset in general, as alignment is inherently based on similarity of data and not the dataset size.
>
> > “The second and third paragraphs in Section 4.4 lack clear logic. They initially suggest that directional alignment is a good predictor of generalization but then recommend using directional dispersion rather than alignment to decide when to stop training.”
>
> Thanks for the feedback, we will focus on improving clarity here.

---

> > ### Comment · Reviewer_Marc · 2024-11-26
> >
> > > We would be grateful if you could elaborate more on what you mean by measuring the effectiveness of GWA.
> >
> > Here I mean analysis of GWA’s effectiveness, i.e., why GWA does better in esitimating generalization than gradient similarity between samples.
> >
> > > Do you have any particular metrics in mind here? As mentioned in line 97, we consider sharpness as one of the most popular alternatives. However, not only does it have shortcomings discussed in Section 2 but we also do not see any additional benefit to using it in comparison to the more informative validation accuracy. We would be happy if you could elaborate more on the potential benefit here.
> >
> > Sharpness estimation (e.g., sensitivity to input noise) and gradient norm are two common ideas. I think comparison with them can be beneficial as they are also simple and popular empirical approaches. For CIFAR-C and CIFAR-P in Table 1, validation accuracy can't be supposed to be an upper bound since the test set follows a different distribution with the training/validation sets.
> >
> > > Are there any specific experiments you would be interested to see? In particular delimited from the results in Table 1.
> >
> > My suggestion is to quantify the correlation between directional alignment and validation accuracy. For example, you can consider to sample multiple checkpoints along the training trajectory and calculate the correlation between the alignment scores and validation accuracy.

---

### Official Review · Reviewer_SoWm · 2024-11-05

**Soundness:** 2
**Presentation:** 3
**Contribution:** 2
**Rating:** 5
**Confidence:** 3

**Summary:**

The paper proposes gradient weight alignment (GWA) as a metric to evaluate the generalization of neural networks during training without requiring a validation set. Specifically, GWA captures the similarity between per-sample gradients and model weights. Furthermore,  directional dispersion (kurtosis of alignment distribution) is used to measure how heavy the tail of alignment score distribution is. A computationally efficient mini-batch estimator for GWA is introduced, making it feasible for large-scale models.

Empirical results show that GWA is a good metric for generalization, and can serve as an early stopping criterion.

**Strengths:**

1. The paper is well-written and easy to follow.

2. The paper conducts extensive experiments to support the claims.

**Weaknesses:**

1. The motivation for using GWA is not clear. Why does a good alignment of training sample gradients indicate good generalization? What if the training samples are noisy and some of the samples may not be useful?

2. It is unclear what properties the lightweight GWA estimator satisfies.

3. Minor

- abstract line 12, "This works advocates" -> "This work advocates"

- lines 408-409, "during in the" -> "during the"

**Questions:**

1. What is the intuition to use alignment between per-sample gradients and the model weights? The former quantifies the change of the weights, which can be very different from the model weights. Is it only suitable for classification models or can also be used in regression models?

2. The paper regards the lightweight GWA estimator to be one of the core contributions. However, it is unclear whether the estimator has desirable properties such as unbiasedness.

3. Why does a good alignment of training sample gradients indicate good generalization? What if the training samples are noisy and some of the samples may not be useful?

---

> ### Author Response · Authors · 2024-11-20
>
> Thank you for reading the paper and reviewing our submission.
>
> > “Why does a good alignment of training sample gradients indicate good generalization?”
>
> We suspect you mean alignment among gradients here. As mentioned in line 130 and following, gradients being similar in their direction across samples indicate that the weights need to change in the same manner for the loss to be reduced for all of these samples. Consequently, the features learned by optimizing with regard to these gradients are general.
>
> > “[...] intuition to use alignment between per-sample gradients and model weights?”
>
> As shown in the work by Ji and Telgarsky (see line 147), the orientation, not the magnitude, of the weights is in theory the key property that describes the prediction capabilities of a model. Intuitively, taking the alignment between the gradient and weights allows us to capture this property, i.e. the change in orientation of the weights required to better predict a specific sample.
>
> > “[...] what if the training samples are noisy [...]”
>
> Uncorrelated sample specific noise is theoretically orthogonal to the direction of general (useful) gradients. In practice, the impact (positive or negative) of noise is highly dependent on the type of noise (see e.g. [1]).  The level of noise is thus reflected by samples being more or less aligned, not only among all gradients as shown in [2] but also with regard to the weight vector (as shown empirically by us in e.g. Section 4.2).
>
> > “[...] unclear what properties the lightweight GWA estimator satisfies.” and “[...] unclear whether the estimator has desirable properties such as being unbiasedness.”
>
> Our estimator is based on the usual assumption of IID sampling and mini-batches being valid empirical estimates of the dataset statistics and the convergence of individual iterates (line 201). We will investigate this formally and will provide a proof when available.
>
> [1] Chen et al., Beyond Class-Conditional Assumption: A Primary Attempt to Combat Instance-Dependent Label Noise, AAAI 2021.
>
> [2] Fort et al., Stiffness: A new perspective on generalization in neural networks, arxiv.org/abs/1901.09491.

---

> > ### Comment · Reviewer_SoWm · 2024-11-25
> > **thanks for the rebuttal**
> >
> > Thanks for the rebuttal, it has addressed most of my concerns.

---

### Meta-Review · Area_Chair_rmwe · 2024-12-07

**Metareview:**

### (a)
This paper studies and proposes a directional alignment measure for deep network training, which is called gradient-weight alignment (GWA). It is the alignment between per-sample gradients and the model weights for each time step such as epoch number or iteration. The paper further proposes to use the kurtosis as a directional dispersion metric to measure the heavy tails of the GWA distribution. The paper proposes a practical version by using a mini batch during training, and uses an exponential moving average approach. Experiments are conducted to show the behavior/performance of the proposed metrics.

### (b)
The proposed method seems to be easy to implement and implementation code is provided. A practical lightweight version is discussed. Various experiments are shown to understand the behavior/performance of the proposed metrics.

### (c)
While the paper presents a new and easy-to-use metric, there were concerns about the significance and novelty of the proposal. By reading the paper, it is somehow challenging to understand the differences and benefits over similar work. For example, the definition of stiffness is shown in Section 4.2, but since this is similar to the proposed metric, it would be better to define this (with equations) in earlier sections, such as Section 2 or Section 3.1. If memory-efficiency is the main contribution along this line, it would be interesting to see how much efficiency gains we have in experiments. Demonstrating these gains could enhance the appeal of the new metric and encourage broader adoption. A reviewer also suggested to add discussions and comparisons with sharpness and gradient norm.

There were also concerns about the position about the paper. A reviewer suggests a direction "to investigate how to use the estimated directional alignment and dispersion to determine an early stopping time without relying on any information from validation sets so that the trained DNN is more robust with respect to different validation sets in the downstream applications." Strengthening this aspect can make the paper even more practical (since the benefit of the proposed method is not that clear in Table 1--it underperforms the validation set baseline in most cases). There is also a paper called "Early Stopping Against Label Noise Without Validation Data" (ICLR 2024) that can be used for comparison.

### (d)
The first paragraph out of the two in (c) was the most important reasoning for the meta-review recommendation.

**Additional Comments On Reviewer Discussion:**

The authors have provided a rebuttal and provided a response to each review. All four reviewers engaged in the discussion with the authors after the rebuttal was provided. Reviewer 383N and Reviewer Marc's points influenced (c) in the meta-review.

---

### Decision · Program_Chairs · 2025-01-22

Reject